# MEMS Electrostatically Driven Coupled Beam Filter Banks

**DOI:** 10.3390/mi14122214

**Published:** 2023-12-07

**Authors:** Richard Syms, Adam Bouchaala

**Affiliations:** Department of Electrical and Electronic Engineering, Imperial College London, Exhibition Road, London SW7 2AZ, UK; a.bouchaala@imperial.ac.uk

**Keywords:** mechanical filter, coupled resonator, MEMS

## Abstract

MEMS bandpass filters based on electrostatically driven, mechanically coupled beams with in-plane motion have been demonstrated up to the VHF band. Filters higher than second order with parallel plate drives have inherent tuning difficulties, which may be resolved by adding mass-loaded beams to the ends of the array. These beams deflect for DC voltages, and thus allow synchronized electrostatic tuning, but do not respond to in-band AC voltages and hence do not interfere with dynamic synchronization. Additional out-of-band responses may be damped, leaving the desired response. The principle is extended here to close-packed banks of filters, with adjacent arrays sharing mass-loaded beams that localize modes to sub-arrays. The operating principles are explained using a lumped element model (LEM) of the equations of motion in terms of resonant modes and the reflection of acoustic waves at discontinuities. Performance is simulated using the LEM and verified using the more realistic stiffness matrix method (SMM) for banks of up to eight filters. Similar or dissimilar filters may be combined in a compact arrangement, and the method may be extended to higher order resonances and alternative coupled resonator systems.

## 1. Introduction

Because gas and material damping can be so low, high Q-factors can be obtained from vacuum packaged, electrostatically driven silicon micro-electro-mechanical systems (MEMS) [1,2,3,4]. Bandpass filters based on coupled resonators with in-plane linear motion and comb and parallel plate drives have been demonstrated at frequencies from tens of kHz [5,6,7,8] to tens of MHz [9,10,11] for intermediate frequency (IF) filtering [7]. Filter banks have also been proposed for channel selection [12,13] and hearing aids [14]. Filters can be electrostatically tuned [15,16,17,18,19], and mechanical coupling between elements can be replaced by electrostatic coupling [20,21,22] or a mixture of the two [23,24]. Methods have been developed to fabricate nanoscale coupling elements [25,26,27,28] and electrode gaps [29,30,31], CMOS integration has been demonstrated [32,33], and alternatives to simple linear arrangements have been proposed [34,35,36,37]. Reviews can be found in [38,39].

The order of the filter depends on the number of coupled resonators, but filters with more than two resonators and parallel plate drives suffer from a conflict between static and dynamic synchronization [40]. We have previously proposed a solution based on the addition of mass-loaded beams at the ends of the array. These deflect for DC voltages, and therefore allow synchronized electrostatic tuning, but do not respond to in-band AC voltages and hence do not interfere with dynamic synchronization. Out-of-band responses generated by these beams may be damped, leaving a good approximation to the desired response. This approach above may be applied to any filter arrangement with a similar conflict. Its main drawback is its inefficient use of wafer area. Here, we extend the principle to close-packed banks of filters, with adjacent arrays sharing mass-loaded beams in a compact arrangement. In this case, the effect of mass loading is mode localization, previously studied as a side effect of disorder [41,42]. With suitable design, banks of filters may be realized for applications such as channel selection.

The structure of this paper is as follows. In Section 2, we explain the principle of mechanically synchronized filters. In Section 3, we review the development of a lumped element model (LEM) of coupled beam systems from perturbation theory and demonstrate how it may be used to simulate performance. In Section 4, we extend the approach to arrays of similar filters, explaining the operation of the composite array in terms of its resonant modes. In Section 5, we provide an alternative explanation in terms of the reflectivity of mass-loaded beams for travelling acoustic waves. In Section 6, we consider arrays of dissimilar filters and show how mode localization allows the formation of filter banks. In Section 7, we confirm performance using the more realistic stiffness matrix method (SMM) and briefly discuss the possibility of the operation on higher order modes. Conclusions are drawn in Section 8.

## 2. Electrostatically Driven Coupled Beam Filters

We start by reviewing the design of a single, mechanically synchronized, electrostatically driven coupled beam filter with parallel plate drives, as described in [40].

### 2.1. Filter Layout

Figure 1a shows a third-order filter, assumed to be fabricated by anisotropic etching and the undercut of a device layer thickness of d. Fixed parts are shown in red and moving parts in blue and cyan, as shown in Figure 1b in a 3D view near the beam roots. The device consists of five suspended built-in beams of length $L_{0}$ and width $w_{0}$ attached to anchors. Each beam is separated from its neighbors by a distance s and connected by a $45^{o}$ meander spring of width $w_{1}$ attached at $x_{1}$ from the beam root. Each beam has its own drive electrode, with an initial electrode gap $g_{0}$. The connections to sources and loads are provided as shown. A DC voltage $V_{D}$ is applied to all electrodes for tuning. Since the electrostatic forces on the beams are equal, each will deflect without deforming the meander spring, so the gap modifies everywhere to g and the resonant frequency is equally shifted.

### 2.2. Mechanical Synchronization

With no electrostatic effects, the inner beams (blue) have identical resonant frequencies based on their own stiffness and that of the meander spring connected on either side. However, because the outer beams are connected to this spring unilaterally, their resonances must be detuned. Although the effect is small, it may spoil a response involving all five beams. Correction may be achieved by modifying the tuning voltages for the outer beams (cyan), but the process is tedious. Instead, these beams are loaded with additional masses at their midpoint, using a short stub connecting each beam to its mass to preserve the design stiffness. Additional electrodes are recessed to allow motion and retain electrostatic uniformity. If the masses are large enough, they can detune these beams so that they cannot participate in collective oscillation near the resonance of the inner beams [40].

The three inner beams comprise the filter, and their number determines its order. Since they behave identically, these beams retain dynamic synchronization. Beam 2 is the input port and is driven by an AC source of voltage $V_{A}$ with output impedance $z_{L}$. Reflection from this port is described by the scattering parameter $S_{11}$. Beam 4 is the output port and terminated by a load of impedance $z_{L}$. Transmission into this port is described by the scattering parameter $S_{21}$. Additional loads are provided to damp the motion of the outer beams; for simplicity, these are also $z_{L}$. We now show that this arrangement can provide a bandpass response.

## 3. Lumped Element Model

In this section, we review our own LEM of a coupled beam filter with a parallel plate drive [36,40], giving the main equations for completeness. This model is based on prior work [9] and has been verified using the SMM and a finite element model (COMSOL^®^ 6).

### 3.1. Modes of a Vibrating Beam

The eigenmodes of an undamped, undriven beam of length $L_{0}$, width $w_{0}$, depth d, density ρ, and Young’s modulus $E_{0}$ can be found from the dynamic Euler equation for a built-in beam [43] as follows:(1)$$\Upsilon_{\nu }\left(x\right)=\left(\frac{\gamma }{\sqrt{L_{0}}}\right)\left\{\frac{sin\left(\beta_{\nu }x\right)-sinh\left(\beta_{\nu }x\right)}{sin\left(\beta_{\nu }L_{0}\right)-sinh\left(\beta_{\nu }L_{0}\right)}-\frac{cos\left(\beta_{\nu }x\right)-cosh\left(\beta_{\nu }x\right)}{cos\left(\beta_{\nu }L_{0}\right)-cosh\left(\beta_{\nu }L_{0}\right)}\right\}$$

Here, $\Upsilon_{\nu }\left(x\right)$ is the transverse displacement and x is the position along the beam. The eigenvalues $\beta_{\nu }$ are related to the resonant frequencies $\omega_{\nu }$ by $\beta_{\nu }^{4}=\omega_{\nu }\rho A_{0}/E_{0}I_{0}$, where $A_{0}=w_{0}d$ is the cross-sectional area of the beam and $I_{0}=w_{0}^{3}d$/12 is its second moment of area. The constant γ is chosen so that $\int_{0}^{L_{0}}{\Upsilon_{\nu }}^{2}dx=1$ and the eigenvalues satisfy the equation $\cos\left(\beta_{\nu }L_{0}\right)\cosh\left(\beta_{\nu }L_{0}\right)=1$. For the lowest order mode, the solution is $\beta_{1}L_{0}=\sqrt{22.37}$. In use, the beam is subjected to a force f and distributed damping r per unit length.

### 3.2. Lumped Element Model

Figure 1c shows the lumped element model in which each resonator except the first and the last is a mass M supported on a spring of stiffness $K_{0}$. Equivalence with the distributed model is established using factors $\eta_{1}=\frac{avg\left(\Upsilon_{1}\right)}{\max\left(\Upsilon_{1}\right)}=0.523$ and $\eta_{2}=\frac{avg\left(\Upsilon_{1}^{2}\right)}{\max\left(\Upsilon_{1}^{2}\right)}=0.396$ which allow M, $K_{0}$, the damping coefficient R, and force F to be found as follows [9]:(2)$$M=\rho A_{0}L_{0}\eta_{2},\;K_{0}=\omega_{1}^{2}M,\;R=rL_{0}\eta_{2},\;F=fL_{0}\eta_{1}$$

For the outer elements, the mass is increased to $m_{r}M$, where $m_{r}=(M+\Delta M)/M$ is a mass ratio and ΔM is an additional mass.

### 3.3. Perturbation Theory

The coupling springs are formed from thinner elements of length $L_{1}$, width $w_{1}$, depth d, density ρ, and Young’s modulus $E_{1}$, inclined at $45^{o}$ to give a separation $s=L_{1}\sqrt{2}$ between the beams. The equivalent spring constant and mass of each pair are $k_{1}=24E_{1}I_{1}/L_{1}^{3}$ and $m_{1}=2\rho A_{1}L_{1}$, where $I_{1}=w_{1}^{3}d/12$ and $A_{1}=w_{1}d$. Here, a different elastic modulus $E_{1}$ is introduced for generality, and the mass $m_{1}$ is half of the actual mass, to model the motion of mass centers. Perturbation theory [44] allows an equivalent lumped element coupling stiffness $K_{1}$ to be found as follows:(3)$$K_{1}=\left(k_{1}-\omega_{1}^{2}m_{1}\right){L_{0}\Upsilon }_{1}^{2}\left(x_{1}\right)\eta_{2}$$

For very small springs, the effect of the mass $m_{1}$ may be ignored, and the coupling stiffness then depends on the strength and positions of the springs.

### 3.4. Electrostatic Transducers

To model electrostatic transducers, we follow [9] and assume that the electrodes act as parallel plate capacitors with capacitance as follows:(4)$${C=\varepsilon }_{0}L_{0}d/\left(g_{0}-y_{D}\right)$$

Here, $g_{0}$ is the initial gap and $y_{D}$ is the static displacement. Application of a DC voltage $V_{D}$ generates a static force as follows:(5)$$F_{D}=\frac{1}{2}C^{\prime }V_{D}^{2}\eta_{1}$$

Here, ${C^{\prime }=\varepsilon }_{0}L_{0}d/{\left(g_{0}-y_{D}\right)}^{2}$ is the derivative of C. Static equilibrium implies that $F_{D}=K_{0e}y_{D}$. Here, $K_{0e}\approx K_{0}$ is the effective stiffness This is a snap-down problem [45], leading to a cubic equation whose solution allows C, $C^{\prime }$, and the second derivative $C^{\prime \prime }$ to be found. If an AC voltage $V_{A}$ is applied from a source with output impedance $z_{L}$, the result is an AC force $F_{A}$, a reduction in stiffness ΔK, and an effective load $Z_{L}$, given by the following:(6)$$F_{A}=V_{D}C^{\prime }\eta_{1}V_{A},\;\Delta K=\frac{1}{2}V_{D}^{2}C\prime \prime \eta_{2},\;Z_{L}=({V_{D}C^{\prime }\eta_{1})}^{2}z_{L}$$

In general, the characteristic impedance $\;Z_{0}$ of a coupled beam array is complex, but for an infinite lossless array at resonance, it has the real value as follows:(7)$$Z_{0R}=K_{1}/\omega_{1e}$$

Here, $\omega_{0}=\surd \left(\frac{k_{0}^{\prime }+2K_{1}}{M}\right)$ is the effective angular resonant frequency, and $K_{0}^{\prime }=K_{0}-\Delta K$. Matching is achieved by choosing $Z_{L}=Z_{0R}$. This requires the load resistance $z_{L}$ to satisfy $K_{v}^{2}z_{L}=Z_{0R}$, where $K_{v}=V_{D}C^{\prime }\eta_{1}$. Very large values of $\;z_{L}$ are needed if $K_{v}$ is small, so electrode gaps must be small to achieve realistic values [9].

### 3.5. Equations of Motion

Early modeling was carried out using equivalent circuits (see e.g., [6,7,8,9]). However, mass loading can be more easily understood with mechanical models. Ignoring the shaded loads, the equations of motion for a five-beam array with ports at n=2 and n=4 and a harmonic drive $F=F_{0}\exp\left(j\omega t\right)$ at angular frequency ω are as follows [40]:$$\left\{K_{0}^{\prime }+K_{1}-m_{r}M\omega^{2}+j\omega R\right\}y_{1}-K_{1}y_{2}=0$$
$$\left\{K_{0}^{\prime }+2K_{1}-M\omega^{2}+j\omega \left(R+Z_{L}\right)\right\}y_{2}-K_{1}(y_{1}+y_{3})=F_{0}$$
$$\left\{K_{0}^{\prime }+2K_{1}-M\omega^{2}+j\omega R\right\}y_{3}-K_{1}(y_{2}+y_{4})=0$$
$$\left\{K_{0}^{\prime }+2K_{1}-M\omega^{2}+j\omega \left(R+Z_{L}\right)\right\}y_{4}-K_{1}(y_{3}+y_{5})=0$$
(8)$$\left\{K_{0}^{\prime }+K_{1}-m_{r}M\omega^{2}+j\omega R\right\}y_{5}-K_{1}y_{4}=0$$

The displacements $y_{n}$ can be found by elimination, and reflection and transmission scattering parameters $S_{11}$ and $S_{21}$ extracted by standard methods.

### 3.6. Example Response

Simulations were carried out for the dimensions in Table 1, which models arrays of weakly coupled beams resonant near 1 MHz. The material parameters in Table 2 were used to model devices in (100) Si with the main and coupling beams in the <110> and <010> directions [46]. A quality factor of $Q=\omega_{0}M/R=5000$ was taken as representative of vacuum packaging; however, its value is unimportant, provided that it is large. A mass ratio $m_{r}=1.5$ was assumed for mass-loaded beams.

The DC voltage $V_{D}$ was first applied to achieve resonance at 1 MHz. An AC voltage of amplitude $V_{A}=0.1\;mV$ was then applied at the input port and the load impedance was adjusted for matching. Figure 2a shows the variation of the S-parameters with frequency. The response is bandpass, with correct tuning and matching at the design frequency (dotted line), but a transmission spike can be seen due to end-beam motion. Figure 2b shows the results with the shaded loads in Figure 1a damping this motion. The response is now purely bandpass, and the dynamic model has reduced to Figure 1d.

## 4. Filter Bank—Similar Filters

In this section, we consider how the mass-loading principle can be extended to a larger array capable of acting as a bank of filters which are initially similar.

### 4.1. Filter Bank

Figure 3a shows an example array containing nine beams, where mass loading (cyan) has been applied to the central beam 5 as well as the end beams 1 and 9. Two sets of three beams (blue) then remain to act as similar third-order filters 1 and 2, separated by a common mass-loaded beam. This arrangement improves wafer utilization by eliminating the need for die separation and (as we show later) can be extended to dissimilar filters. The input and output ports of filter 1 are 1 and 2, while those of filter 2 are 3 and 4.

Electrical connections are omitted for simplicity but are analogous to those in Figure 1a. DC voltages $V_{D}$ must be applied to each electrode and AC voltages $V_{A1}$ and $V_{A2}$ to the two input ports 1 and 3, using sources with impedance $z_{L}$. Similarly, the two output ports 2 and 4 must be connected to loads, and the mass-loaded beams should again be damped. The scattering parameters of filter 1 are $S_{11}$ and $S_{21}$, while those of filter 2 are $S_{33}$ and $S_{43}$. The design supports global electrostatic tuning but not the tuning of individual filters; this would involve compression or extension of the meander springs near the central beam, which would then be relaxed by the springs elsewhere.

### 4.2. Equations of Motion

The lumped element equivalent of this arrangement is shown in Figure 3b. Omitting damping on the mass-loaded beams to begin with, the equations of motion for harmonic drives $F_{A}=F_{0A}\exp\left(j\omega t\right)$ and $F_{B}=F_{0B}\exp\left(j\omega t\right)$ are as follows:$$\left\{K_{0}^{\prime }+K_{1}-m_{r}M\omega^{2}+j\omega R\right\}y_{1}-K_{1}y_{2}=0$$
$$\left\{K_{0}^{\prime }+{2K}_{1}-M\omega^{2}+j\omega \left(R+Z_{L}\right)\right\}y_{2}-K_{1}(y_{1}+y_{3})=F_{0A}$$
$$\left\{K_{0}^{\prime }+{2K}_{1}-M\omega^{2}+j\omega R\right\}y_{3}-K_{1}(y_{2}+y_{4})=0$$
$$\left\{K_{0}^{\prime }+{2K}_{1}-M\omega^{2}+j\omega \left(R+Z_{L}\right)\right\}y_{4}-K_{1}(y_{3}+y_{5})=0$$
$$\left\{K_{0}^{\prime }+{2K}_{1}-m_{r}M\omega^{2}+j\omega R\right\}y_{5}-K_{1}(y_{4}+y_{6})=0$$
$$\left\{K_{0}^{\prime }+{2K}_{1}-M\omega^{2}+j\omega \left(R+Z_{L}\right)\right\}y_{6}-K_{1}(y_{5}+y_{7})=F_{0B}$$
$$\left\{K_{0}^{\prime }+{2K}_{1}-M\omega^{2}+j\omega R\right\}y_{7}-K_{1}(y_{6}+y_{8})=0$$
$$\left\{K_{0}^{\prime }+{2K}_{1}-M\omega^{2}+j\omega \left(R+Z_{L}\right)\right\}y_{8}-K_{1}(y_{7}+y_{9})=0$$
(9)$$\left\{K_{0}^{\prime }+K_{1}-m_{r}M\omega^{2}+j\omega R\right\}y_{9}-K_{1}y_{8}=0$$

These equations can be solved as before, but we first focus on the resonant modes.

### 4.3. Eigenmodes

In the absence of damping and driving forces, Equation (9) reduces to the following:$$(\omega_{2}^{2}-\omega^{2})y_{1}-\kappa_{m}y_{2}=0$$
$$(\omega_{0}^{2}-\omega^{2})y_{2}-\kappa (y_{1}+y_{3})=0$$
$$(\omega_{0}^{2}-\omega^{2})y_{3}-\kappa (y_{2}+y_{4})=0$$
$$(\omega_{0}^{2}-\omega^{2})y_{4}-\kappa (y_{3}+y_{5})=0$$
$$(\omega_{1}^{2}-\omega^{2})y_{5}-\kappa_{m}(y_{4}+y_{6})=0$$
$$(\omega_{0}^{2}-\omega^{2})y_{6}-\kappa (y_{5}+y_{7})=0$$
$$(\omega_{0}^{2}-\omega^{2})y_{7}-\kappa (y_{6}+y_{8})=0$$
$$(\omega_{0}^{2}-\omega^{2})y_{8}-\kappa (y_{7}+y_{9})=0$$
(10)$$(\omega_{2}^{2}-\omega^{2})y_{9}-\kappa_{m}y_{8}=0$$

Here, further resonant frequencies are defined as $\omega_{1}^{2}=(K_{0}^{\prime }+{2K}_{1})/m_{r}M$ and $\omega_{2}^{2}=(K_{0}^{\prime }+K_{1})/m_{r}M$, and new coupling terms are given by $\kappa =K_{1}/M$ and $\kappa_{m}=K_{1}/(m_{r}M)$.

For characteristic modes oscillating at the μth angular resonant frequency $\omega_{\mu }$, we may write $y_{\mu n}=Y_{\mu n}\;exp(j\omega_{\mu }t)$, where the constants $Y_{\mu n}$ define the overall mode shapes. The resonant frequencies are the eigenvalues of the tridiagonal matrix $\underline{M}$, given by the following:(11)$$\underline{M}=\left(\begin{array}{c} \begin{array}{ccc} \omega_{2}^{2} & -\kappa_{m} & \begin{array}{ccc} 0 & 0 & \begin{array}{ccc} 0 & 0 & \begin{array}{ccc} 0 & 0 & 0 \end{array} \end{array} \end{array} \end{array}\\ \begin{array}{ccc} -\kappa & \omega_{0}^{2} & \begin{array}{ccc} -\kappa & 0 & \begin{array}{ccc} 0 & 0 & \begin{array}{ccc} 0 & 0 & 0 \end{array} \end{array} \end{array} \end{array}\\ \begin{array}{c} \begin{array}{ccc} 0 & -\kappa & \begin{array}{ccc} \omega_{0}^{2} & -\kappa & \begin{array}{ccc} 0 & 0 & \begin{array}{ccc} 0 & 0 & 0 \end{array} \end{array} \end{array} \end{array}\\ \begin{array}{ccc} 0 & 0 & \begin{array}{ccc} -\kappa & \omega_{0}^{2} & \begin{array}{ccc} -\kappa & 0 & \begin{array}{ccc} 0 & 0 & 0 \end{array} \end{array} \end{array} \end{array}\\ \begin{array}{c} \begin{array}{ccc} 0 & 0 & \begin{array}{ccc} 0 & -\kappa_{m} & \begin{array}{ccc} \omega_{1}^{2} & -\kappa_{m} & \begin{array}{ccc} 0 & 0 & 0 \end{array} \end{array} \end{array} \end{array}\\ \begin{array}{ccc} 0 & 0 & \begin{array}{ccc} 0 & 0 & \begin{array}{ccc} -\kappa & \omega_{0}^{2} & \begin{array}{ccc} -\kappa & 0 & 0 \end{array} \end{array} \end{array} \end{array}\\ \begin{array}{c} \begin{array}{ccc} 0 & 0 & \begin{array}{ccc} 0 & 0 & \begin{array}{ccc} 0 & -\kappa & \begin{array}{ccc} \omega_{0}^{2} & -\kappa & 0 \end{array} \end{array} \end{array} \end{array}\\ \begin{array}{ccc} 0 & 0 & \begin{array}{ccc} 0 & 0 & \begin{array}{ccc} 0 & 0 & \begin{array}{ccc} -\kappa & \omega_{0}^{2} & -\kappa \end{array} \end{array} \end{array} \end{array}\\ \begin{array}{ccc} 0 & 0 & \begin{array}{ccc} 0 & 0 & \begin{array}{ccc} 0 & 0 & \begin{array}{ccc} 0 & -\kappa_{m} & \omega_{2}^{2} \end{array} \end{array} \end{array} \end{array} \end{array} \end{array} \end{array} \end{array}\right)$$

Figure 4a shows the variation of the normalized frequencies $\omega_{\mu }/\omega_{0}$ with the mass ratio $m_{r}$ for the example stiffness ratio $K_{1}/K_{0}^{\prime }=0.0244$ obtained using the parameters of Table 1 and Table 2. The results fall into two groups, each containing distinct values when $m_{r}=0$. The upper band (blue) represents modes involving motion of the filter beams. The width of this band depends on $K_{1}/K_{0}^{\prime }$, which ultimately determines the filter bandwidth. For typical applications, this ratio must be small, implying that the width of the coupling spring must be small compared with that of the main beams. As $m_{r}$ increases, each pair of resonances degenerates to a constant value. The lower group (cyan) contains modes of the mass-loaded beams. As $m_{r}$ increases, two of these resonances (symmetric and antisymmetric modes of the end beams) again degenerate, while the third (involving the central beam) remains slightly higher and the group separates further from the main band.

Figure 4b shows the corresponding mode shapes for $m_{r}=1.5$. The two upper plots show that the blue resonances are collective symmetric and antisymmetric modes with zeros at the ends and center of the array. In this case, there is no mode localization, and the injection of a signal into either filter must excite both sets of modes together. However, the modes may then add in one filter and cancel in the other to give the appearance of excitation of a single filter. Numerical analysis shows that effective cancellation merely requires a sufficiently large value of $m_{r}$. The lower plot confirms that the cyan resonances in Figure 4a involve the loaded beams alone. Consequently, for large $m_{r}$, there will be no motion of these beams except near discrete out-of-band frequencies.

### 4.4. Example Response

Figure 5a shows the frequency variation of the S-parameters of filter 1 for the parameters of the previous section and $m_{r}=1.5$, assuming that the mass-loaded beams are damped. The response is essentially that of Figure 2b, a bandpass response. The responses obtained when filter 2 is excited are similar, implying that independent operation of the two filters has been achieved. However, because the filters are coupled together, there is some potential for crosstalk. Figure 5b shows the frequency variation of the outputs from filter 2. The unwanted outputs are everywhere below ~−30 dB and can be suppressed further by increasing $m_{r}$. However, it should be noted that there are limits on the achievable value of $m_{r}$ due to the finite space occupied by the loading elements.

## 5. Mass-Loaded Beams as Reflectors

We now provide an explanation for the subdivision of the array by considering mass-loaded beams as reflectors for travelling acoustic waves.

### 5.1. Dispersion Equation for Acoustic Waves

We first consider an infinite line of identical coupled beams with resonant frequency $\omega_{0}$ and coupling constant κ. In the absence of loss, the equations of motion are as follows:(12)$$(\omega_{0}^{2}-\omega^{2})y_{n}-\kappa (y_{n-1}+y_{n+1})=0$$

In the uniform regions, we may assume travelling wave solutions in the following form:(13)$$y_{n}=y_{0}exp\left(-jnka\right)$$

Here, $y_{0}$ is the wave amplitude, k is the propagation constant, and a is the spacing between resonators. Substituting into Equation (14), we obtain the following dispersion equation:(14)$${\omega^{2}=\omega }_{0}^{2}-2\kappa \;cos\left(ka\right)$$

Figure 6a shows the dispersion characteristic, assuming $K_{1}/K_{0}=0.0244$ as before. The variation is essentially that of an acoustic slow-wave structure, and propagation is allowed only over a finite band of frequencies. The addition of loss will introduce a complex propagation constant, so waves decay as they propagate, and allow out-of-band propagation near the band edges; however, the effect will be small with high Q-factors.

### 5.2. Transmission and Reflection at a Discontinuity

We now assume that there is a perturbation in one beam, which for simplicity we take as beam zero as in Figure 3c. Here, the resonant frequency and coupling are modified to $\omega_{1}=\omega_{0}/\surd m_{r}$ and $\kappa_{m}=\kappa /m_{r}$. Near this point, the equations modify to the following:$$(\omega_{0}^{2}-\omega^{2})y_{-1}-\kappa (y_{-2}+y_{0})=0$$
(15)$$\left(\omega_{1}^{2}-\omega^{2}\right)y_{0}-\kappa_{m}\left(y_{-1}+y_{+1}\right)=0$$

We assume that an upward travelling wave is incident on the discontinuity. Standard physics suggests that this wave will be scattered into reflected and transmitted waves, so we assume solutions in the form of appropriate travelling wave terms:$$y_{n}=y_{I}exp\left(-jnka\right)+y_{R}exp\left(+jnka\right)\;\;\;\;\;\;\;\;\;\;\;\;\;\;\;\;\;\;\;n<0$$
(16)$$\;\;\;\;\;\;\;\;\;\;\;\;\;\;\;\;\;\;\;\;y_{n}=y_{T}exp\left(-jnka\right)\;\;\;\;\;\;\;\;\;\;\;\;\;\;\;\;\;\;\;\;\;\;\;\;\;\;\;\;\;\;\;\;\;n\geq 0$$

Here, $y_{I}$, $y_{R}$, and $y_{T}$ are the amplitudes of the incident reflected and transmitted waves, respectively. Substituting into Equation (15) and making use of Equation (14), the reflection and transmission coefficients $R=y_{R}/y_{I}$ and $T=y_{T}/y_{I}$ can be found as follows:(17)$$R=-\frac{\omega_{1}^{2}-\omega_{0}^{2}+2\left(\kappa -\kappa_{m}\right)\;cos\left(ka\right)}{\omega_{1}^{2}-\omega_{0}^{2}+2\left(\kappa -\kappa_{m}\right)\;cos\left(ka\right)+2j\kappa_{m}sin\left(ka\right)}$$
(18)$$T=\frac{2j\kappa_{m}sin\left(ka\right)}{\omega_{1}^{2}-\omega_{0}^{2}+2\left(\kappa -\kappa_{m}\right)\;cos\left(ka\right)+2j\kappa_{m}sin\left(ka\right)}$$

Since ${\left|R\right|}^{2}+{\left|T\right|}^{2}=1$, these equations conserve power, and they satisfy the standard relations T=1+R and reduce to R=0 and T=1 when $m_{r}=1$. Figure 6b shows the variation of $\left|R\right|$ for the same parameters as Figure 6a and different values of $m_{r}$. The reflectivity varies across the band, with R=−1 at the band edges when ka=0 or ka=π and the smallest effect near the band center. However, as $m_{r}$ rises, R→−1 over the entire band and the loaded beam acts as a mirror for acoustic waves. Thus, mass loading can be understood in terms of reflectors that divide an array into independent sub-arrays.

## 6. Filter Bank—Dissimilar Filters

We now consider how mass loading can create banks of dissimilar filters, differing in their order or their center frequency (or both). In either case, mode localization occurs.

### 6.1. Dissimilar Orders

For example, Figure 7a shows a 10-beam array subdivided into third-order (blue) and fourth-order (green) filters using mass-loaded beams (cyan). The coupled equations can be developed as before and then reduced to an eigenvalue–eigenvector problem.

Figure 8a shows the variation of the normalized resonant frequencies $\omega_{\mu }/\omega_{0}$ with $m_{r}$ for the previous parameters. The filter resonances again stabilise as $m_{r}$ increases but now form superimposed three-mode (blue) and four-mode (green) bands. A further group (cyan) involving the end beams again gradually diverges. Figure 8b shows the mode shapes for $m_{r}=1.5$, showing that the main modes are now localized to the two filters. In contrast to the previous section, the operation of a single filter now involves a subset of modes, and no cancellation is involved. Figure 9a,b show the frequency dependence of the S-parameters for the two filters. In each case, tuning and matching are correct and a bandpass response is obtained; however, the filter responses are clearly of different order.

### 6.2. Dissimilar Frequencies

The principle may be extended to dissimilar center frequencies. For example, Figure 7b shows a nine-element array subdivided into two third-order filters (blue and green) using mass-loaded beams (cyan). The port arrangements are unchanged, but the filter elements are now also mass loaded to tune their resonant frequency, using equivalent masses $m_{r1}M$ and $m_{r2}M$. If the two mass ratios are written in the form $m_{ri}=1+{∆m}_{ri}$, we would expect the corresponding center frequencies $\omega_{0i}$ to scale as follows:(19)$$\omega_{0i}\propto \frac{1}{\sqrt{m_{ri}}}\approx 1-\frac{{∆m}_{ri}}{2}$$

This approximation allows suitable values of $m_{ri}$ to be estimated; further correction can then be used to space the center frequencies more accurately. Figure 10a shows the variation of the normalized resonant frequencies $\omega_{\mu }/\omega_{0}$ with $m_{r}$, assuming the same parameters and $m_{r1}=1,\;m_{r2}=1.1$, for which the normalised resonant frequencies can be estimated as 1 and 0.95. The resonances now fall into separate bands corresponding to motion in filters 1 (blue) and 2 (green) and a further group involving the end beams (cyan). As before, the filter resonances stabilise when $m_{r}$ is large. Figure 10b shows the mode shapes with $m_{r}=1.5$ which confirm that the main modes are again localized.

Figure 11a shows the frequency dependence of transmission for each filter, with $V_{D}$ adjusted to center the response around 1 MHz and to the load resistors altered to improve matching; these must be scaled as ${z_{Li}=z}_{L}\sqrt{m_{ri}}$. Here, ‘1’ denotes the transmission of filter 1, and so on. In each case, a bandpass response is obtained, but the center frequencies are up- and down-shifted, implying that the array has provided two different filters.

### 6.3. Filter Banks

The unoccupied frequency interval between the green and cyan resonances in Figure 10a suggests that the mass-loading principle may be extended to larger filter banks. This is indeed the case, as all that is required is that the center frequencies are equally spaced and the bands do not overlap with each other or with the resonances of the mass-loaded beams. At minimum, this requires that mass ratio $m_{ri}$ of the ith filter be chosen so that $m_{r1}=1$, $m_{ri}>m_{ri-1}$, and $m_{ri}<m_{r}$. For the parameters here, four filters may be realised. Figure 11b shows the results obtained by mass loading the arrays so that $m_{ri}=1+0.1(i-1)$ and $m_{r}=1.5$, and again adjusting $V_{D}$ and load resistors to center the response and improve matching. A set of four near-identical frequency-shifted responses is now obtained in the format needed for a channel selector or multiplexer.

The maximum number of channels depends on the filter bandwidth. For the parameters here, similar results can be obtained for eight filters provided that their bandwidth is reduced to avoid channel overlap by weakening the coupling springs, choosing suitable mass ratios, and adjusting tuning and matching. Figure 11c shows the results obtained by increasing the separation s to 10 μm and using the mass ratios $m_{ri}=1+0.05(i-1)\;and\;m_{r}=1.5$. Eight near-identical responses are obtained. The overall bandwidth may itself be raised by increasing $m_{r}$. In practice, there are limits set by the physical size of the masses; we discuss this further in Section 7.

## 7. Stiffness Matrix Method

Distributed devices can be simulated using the finite element method (FEM) [47] and the stiffness matrix method [48]. The FEM solves partial differential equations with boundary conditions by subdividing space into a mesh and reducing the problem to a system of nodal equations. It can provide accurate models of MEMS involving different physical domains, but calculations are extremely lengthy for high-aspect-ratio sub-spaces when the number of nodes is large [49]. The SMM replaces flexible elements with equivalent stiffness terms from the Euler beam bending theory [43], and electrostatic transducers with analytic approximations [50]. The result is again simultaneous equations, but their number is greatly reduced. Both methods allow the exploration of realistic layouts, without assuming responses involve specific modes. To avoid excessive run times associated with large coupled-beam arrays using FEM, we focus here on the SMM.

### 7.1. SMM Solver

Two-dimensional SMM calculations were performed using a Matlab^®^ R2020a model previously verified against FEM [36,40]. The stiffness matrix K was constructed from the layout, dimensions, and material parameters, with $E_{0}$ reduced to model electrostatic detuning. Long beams were divided into 16 sections to improve accuracy. Axial, transverse, and angular displacements at nodes were found for a vector of applied forces and torques. 

Dynamic analysis was performed using additional mass and damping matrices. The mass matrix M was formed by combining dimensions and densities with standard relations for motions of centers of mass. The damping matrix C was modeled using Rayleigh’s method as R=aM+bK [43]. Here, a and b are mass and spring damping coefficients, with a determined from the Q-factor and b set to zero. Ports were simulated by increasing the damping for these beams, using a damping coefficient determined from the load impedance. Assuming harmonic forces and displacements as $\left(F,U\right)\;e^{j\omega t}$, substitution into the governing equation yields $\left(K-\omega^{2}M+j\omega R\right)U=F$. This equation was solved by matrix inversion, and the velocity vector was constructed as S=jωU. The scattering parameters were then extracted from midpoint velocities.

### 7.2. Physical Layouts

Simulations were carried out for the parameters in Table 1 and Table 2. For example, Figure 12a shows the statically deflected shape of an array containing two unloaded three-element sub-arrays, as in Figure 3a. Here, thick horizontal lines represent the main beams and thin inclined lines represent the meander springs. Anchors and drive electrodes are not shown. The beams are loaded with masses on elements 1, 5, and 9. The short stub connecting each beam to its mass is omitted, as it will have no effect in a SMM model.

The physical size of the masses illustrates the main limitation. Here, a mass ratio $m_{r}=1.5$ is achieved using a mass measuring 4 μm×22.5 μm; larger masses will consume a greater fraction of the space allocated to electrodes, and care will be needed to ensure electrostatic uniformity. The beams deflect identically as expected.

### 7.3. Mode Synchronization

Figure 13a shows the variation of the normalized eigenfrequencies with mass ratio $m_{r}$. This figure should be compared with earlier results from the LEM in Figure 4a; the qualitative agreement is excellent, and the modes synchronize correctly as $m_{r}$ rises. Minor differences can be ascribed to the approximations inherent in perturbation theory. Figure 13b shows a similar variation over an enlarged frequency range to illustrate the effect of higher order modes. The resonances of the first-order (symmetric) modes lie at the bottom of the plot. A second group of resonances lie at the center; these correspond to second-order (antisymmetric) modes. Since their mode shapes have zeros at the point of mass loading, there can be no variation in their resonances with $m_{r}$. A third group lie at the top; these involve third-order symmetric modes. This group is also synchronized as $m_{r}$ rises. These results confirm that basic behaviour is captured well by the LEM but admit the possibility of operating on higher order symmetric modes.

### 7.4. Filter Banks

Responses predicted by the LEM were verified by the SMM. All were qualitatively identical, so we give only limited examples. Mass ratios $m_{ri}$ corresponding to those of Figure 11 were obtained by assuming a constant mass width of 22.5 μm and scaling the mass depths appropriately. Array layouts were predictable; for example, Figure 12b shows the deflected shape of an array containing four mass-loaded filters. Figure 14 shows the frequency dependence of transmission for banks of (a) 2, (b) 4, and 8 filters, with beam separations of s=8 μm (a, b) and s=10 μm (c). Calculations involved 186, 346, and 666 nodes, respectively, and run times were ~10, 60, and 600 s using a 2.6 GHz MacBook Pro, enabling practical exploration of design spaces. This figure should be compared with earlier results from the LEM in Figure 11; there is excellent agreement, confirming that ideal performance can be approached using devices with realizable dimensions.

Numerically, we have shown that channel isolation can be improved by raising the number of beams in each array and hence the order of each filter. We have also verified that mode localization and the formation of filter banks can be achieved using third-order modes if the mass ratios, tuning, and matching are modified appropriately. This observation might allow frequencies to be raised without further dimensional reductions.

## 8. Conclusions

Mass loading has been investigated as a method of forming filter banks based on arrays of coupled beams with parallel plate electrostatic drives. Its effect is to render loaded beams stationary at excitation frequencies within the operating band. For isolated filters, it can provide terminating elements that permit static deflection while suppressing dynamic motion. However, mass loading can also allow intermediate elements to act as reflectors that localize modes to sub-arrays. This principle can be used to develop banks of filters with different center frequencies for channel selection and multiplexing. The architecture is scalable using simple design rules and the reduction in die size following on from beam sharing improves wafer utilization. Principles have been established using a lumped element model and confirmed using the stiffness matrix method. The main difference is the appearance of higher order modes in the SMM, but similar effects occur at these higher frequency bands and may be exploitable in device applications.

The designs investigated have several performance limitations. The space available for mass-loading elements is small, limiting desynchronization and restricting overall bandwidth. The replacement of built-in beams with cantilevers would allow larger masses to be placed at beam ends. The use of continuous anchors renders any electrostatic tuning global. Although this allows for compensation of global dimensional errors, anchor segmentation would allow for the tuning of individual filters.

There are additional fabrication challenges. The width $w_{1}$ of the meander springs must be much less than the width $w_{0}$ of the main beams so that the springs are a minor perturbation. The electrode gap $g_{0}$ must also be small to avoid unrealistic characteristic impedance. Consequently, nanofabrication will be required to define the layout. Formation of a deep etched structure will then require high sidewall verticality. Connection to internal electrodes might require flip-chip bonding to an electrical backplane.

Some potential solutions exist. The principle is likely applicable to electrostatically coupled resonators, avoiding the problem of fabricating nanoscale features together with nanoscale gaps. The replacement of built-in beams with cantilevers would simplify the connection to electrodes, which may then be accessed from the edge of the array. The principle is also likely to be applicable to torsional resonators, using additional rotary inertia for loading. The coupling elements and electrode gaps will then have nanoscale out-of-plane dimensions, which may be defined as layer thicknesses rather than lithography. The ability to place loading elements above rather than between resonators may also increase achievable loading effects. These alternatives are under investigation.

## Figures and Tables

**Figure 1 micromachines-14-02214-f001:**
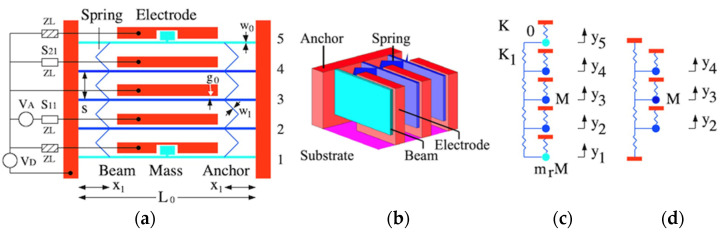
Synchronized third-order filter: (**a**) layout; (**b**) 3D view; (**c**,**d**) static and dynamic LEMs.

**Figure 2 micromachines-14-02214-f002:**
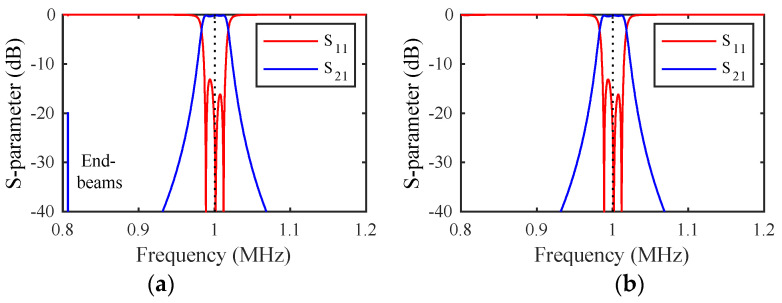
Frequency response of a third-order filter (**a**) without and (**b**) with end-beam damping. LEM.

**Figure 3 micromachines-14-02214-f003:**
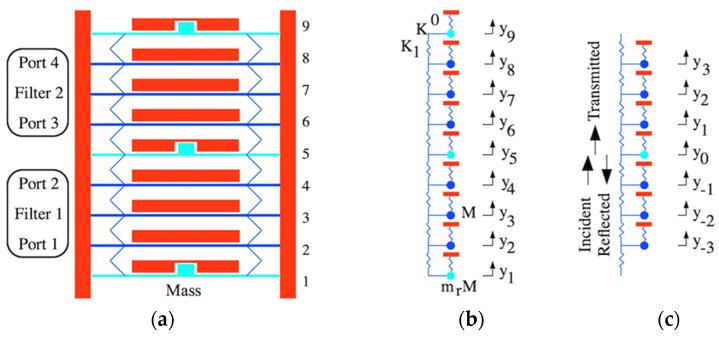
Bank of identical third-order filters: (**a**) layout; (**b**) LEM; (**c**) LEM for infinite array with defect.

**Figure 4 micromachines-14-02214-f004:**
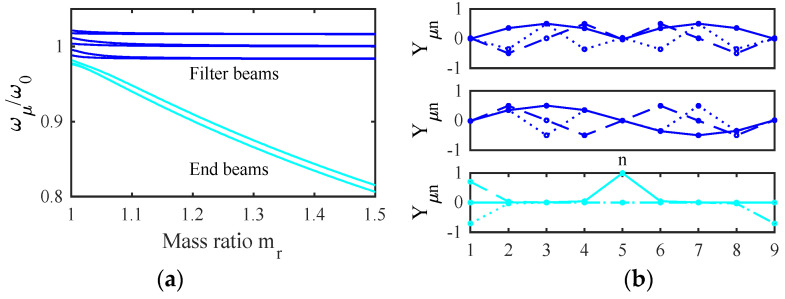
A two-filter bank: (**a**) variation of resonant frequency with $m_{r}$ (**b**) mode shapes with $m_{r}=1.5$.

**Figure 5 micromachines-14-02214-f005:**
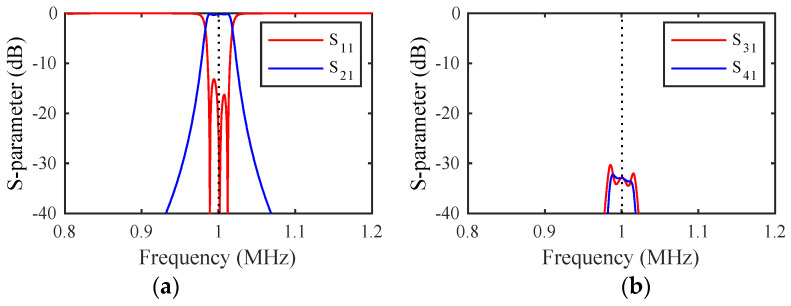
S-parameter variations with filter 1 excited: outputs from (**a**) filter 1 and (**b**) filter 2. LEM.

**Figure 6 micromachines-14-02214-f006:**
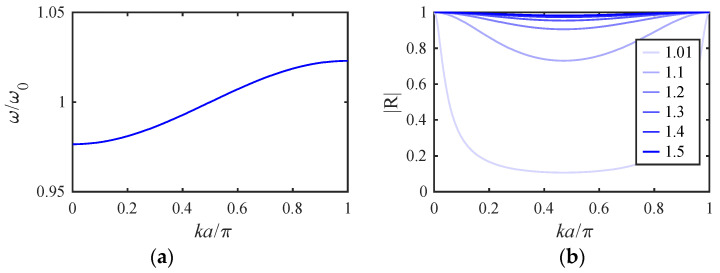
Infinite array: (**a**) dispersion and (**b**) variation of |R| with ka at reflectors with different $m_{r}$.

**Figure 7 micromachines-14-02214-f007:**
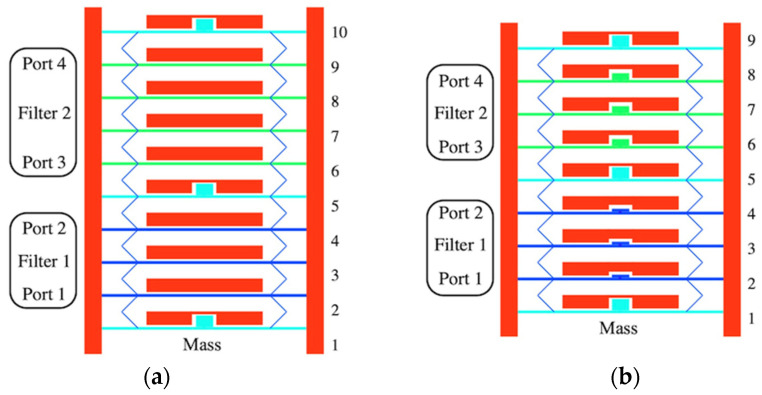
Banks of non-identical filters: (**a**) dissimilar order and (**b**) dissimilar frequency.

**Figure 8 micromachines-14-02214-f008:**
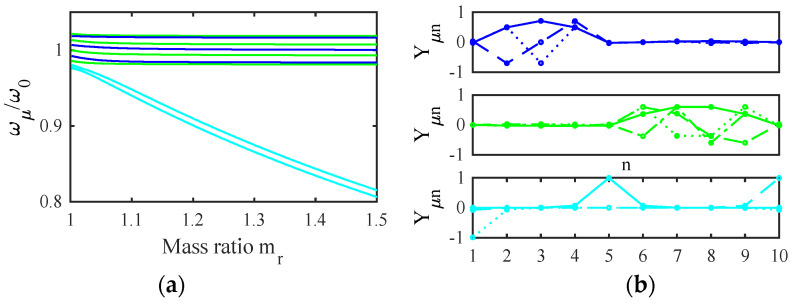
Filters of dissimilar order: (**a**) variation of $\omega_{\mu }$ with $m_{r}$; (**b**) eigenmodes for $m_{r}=1.5$.

**Figure 9 micromachines-14-02214-f009:**
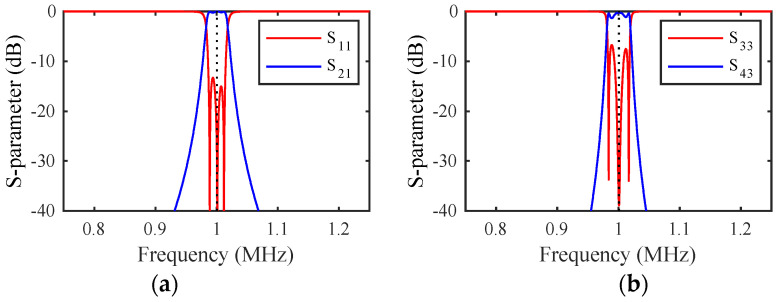
Filters of dissimilar order: response of (**a**) filter 1 and (**b**) filter 2. LEM.

**Figure 10 micromachines-14-02214-f010:**
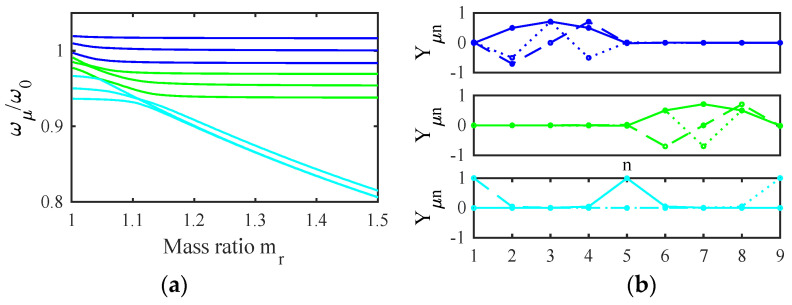
Filters of dissimilar frequency: (**a**) variation of $\omega_{\mu }$ with $m_{r}$; (**b**) eigenmodes for $m_{r}=1.5$.

**Figure 11 micromachines-14-02214-f011:**
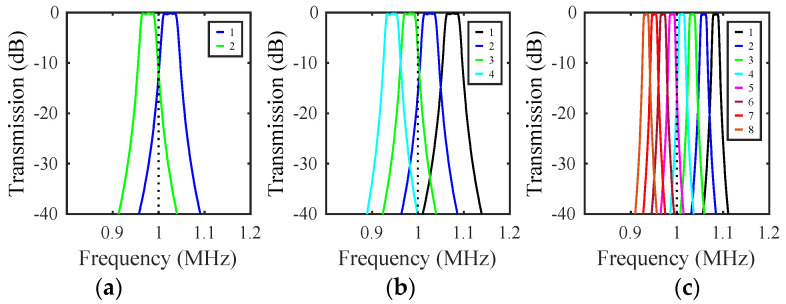
Frequency dependence of transmission for banks of (**a**) 2, (**b**) 4, and (**c**) 8 filters. LEM.

**Figure 12 micromachines-14-02214-f012:**
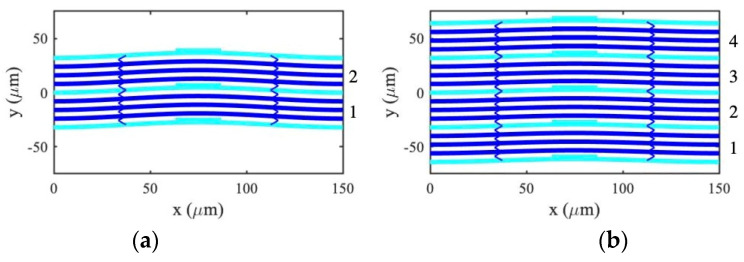
Layout and deflected shape of banks of (**a**) 2 and (**b**) 4 filters. SMM.

**Figure 13 micromachines-14-02214-f013:**
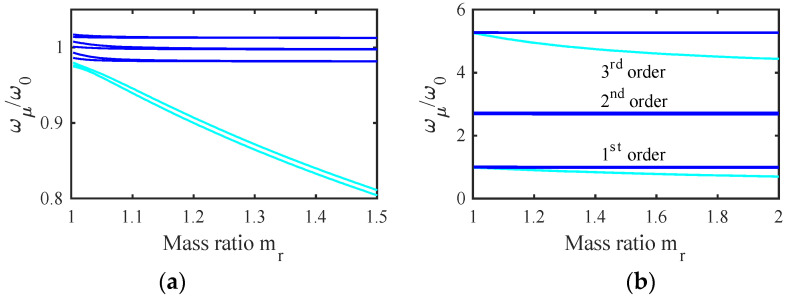
Bank of similar filters: variation of resonant frequencies with $m_{r}$ for (**a**) lowest order group and (**b**) lowest three groups. SMM.

**Figure 14 micromachines-14-02214-f014:**
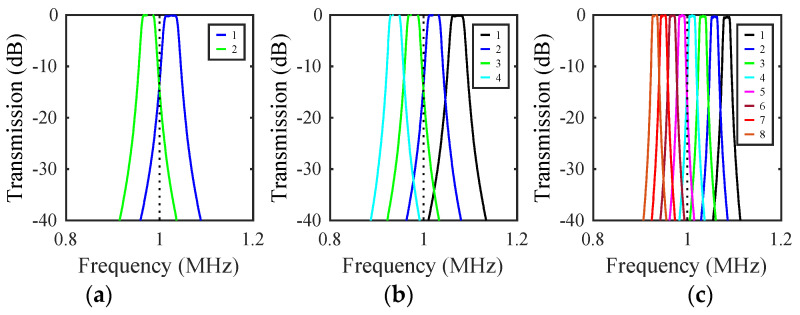
Frequency dependence of transmission for banks of (**a**) 2, (**b**) 4, and (**c**) 8 filters. SMM.

**Table 1 micromachines-14-02214-t001:** Layout parameters used in the LEM simulation.

$L_{0}$	$w_{0}$	d	$\alpha =x_{1}/L_{0}$	s	$w_{1}$	$g_{0}$
$\left(\mu m\right)$	$\left(\mu m\right)$	$\left(\mu m\right)$		$\left(\mu m\right)$	$\left(\mu m\right)$	$\left(\mu m\right)$
150	3	4	0.25	8	0.1	0.1

**Table 2 micromachines-14-02214-t002:** Material and other parameters used in the LEM simulation.

ρ	$E_{0}$	$E_{1}$	$V_{D}$	$z_{L}$	Q	$m_{r}$
$\left(kg/m^{3}\right)$	$\left(N/m^{2}\right)$	$\left(N/m^{2}\right)$	$\left(V\right)$	$\left(k\Omega \right)$		
2332	$169\times 10^{9}$	$130\times 10^{9}$	2.72	207	5000	1.5

## Data Availability

Data are contained within the article.

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
