# Peer review of "MEMS Electrostatically Driven Coupled Beam Filter Banks"

_micromachines, 2023, doi:10.3390/mi14122214_

Round 1

Reviewer 1 Report

Comments and Suggestions for Authors

The work looks novel and of interest to the scientific community; moreover it is quite clearly presented. The theoretical aspects are well developed. From my point of view the physical structure of the filter bank should be better described using an additional figure or by improving figure 1. Currently it is not easy to understand how the suspended and fixed parts of the device are arranged. This would significantly improve the comprehensibility of its operation. If the authors had a photo of a real device, it could clarify many doubts about its operation.

Author Response

The work looks novel and of interest to the scientific community; moreover it is quite clearly presented. The theoretical aspects are well developed.

Thank you for these kind comments.

From my point of view the physical structure of the filter bank should be better described using an additional figure or by improving figure 1. Currently it is not easy to understand how the suspended and fixed parts of the device are arranged. This would significantly improve the comprehensibility of its operation. If the authors had a photo of a real device, it could clarify many doubts about its operation.

Thank you for this comment. We have added an additional sub-figure to Figure 1 to show the proposed 3D structure of the device as requested.

Reviewer 2 Report

Comments and Suggestions for Authors

Your article is well-structured and full of rich content, however, there is a minor issue that need to be addressed.

1.      The second section of your article primarily references your previously published work. It would be beneficial to explicitly state in the text that this is a result of your prior research. Both the second and third sections are quite lengthy and heavily cite your own literature. It would be advisable to condense these sections to highlight the innovative content of this paper.

Author Response

Your article is well-structured and full of rich content.

Thank you for these kind comments.

However, there is a minor issue that need to be addressed.

  1. The second section of your article primarily references your previously published work. It would be beneficial to explicitly state in the text that this is a result of your prior research.

We have explicitly stated this aspect as requested.

  1. Both the second and third sections are quite lengthy and heavily cite your own literature. It would be advisable to condense these sections to highlight the innovative content of this paper.

We appreciate this comment. However, the present MS relies heavily on our own past work and is indeed a development of it, so some background is needed for understanding. We have therefore condensed these sections rather than eliminate them. Since the changes mainly involve deletion, they are not obvious from the marked-up text, but we have removed 11 lines (including 1 equation), adjusting text elsewhere to fit. Ignoring the introduction and references, background material and new research now involve ca 3 and 9 pages, respectively.